# Beneficial Effects of Hyaluronan-Based Hydrogel Implantation after Cortical Traumatic Injury

**DOI:** 10.3390/cells11233831

**Published:** 2022-11-29

**Authors:** Anaïs Lainé, Sébastien Brot, Afsaneh Gaillard

**Affiliations:** Laboratory of Experimental and Clinical Neurosciences, INSERM U-1084, University of Poitiers, CEDEX 09, 86073 Poitiers, France

**Keywords:** biomaterial, hyaluronan, traumatic brain injury, neuroinflammation

## Abstract

Traumatic brain injury (TBI) causes cell death mainly in the cerebral cortex. We have previously reported that transplantation of embryonic cortical neurons immediately after cortical injury allows the anatomical reconstruction of injured pathways and that a delay between cortical injury and cell transplantation can partially improve transplantation efficiency. Biomaterials supporting repair processes in combination with cell transplantation are in development. Hyaluronic acid (HA) hydrogel has attracted increasing interest in the field of tissue engineering due to its attractive biological properties. However, before combining the cell with the HA hydrogel for transplantation, it is important to know the effects of the implanted hydrogel alone. Here, we investigated the therapeutic effect of HA on host tissue after a cortical trauma. For this, we implanted HA hydrogel into the lesioned motor cortex of adult mice immediately or one week after a lesion. Our results show the vascularization of the implanted hydrogel. At one month after HA implantation, we observed a reduction in the glial scar around the lesion and the presence of the newly generated oligodendrocytes, immature and mature neurons within the hydrogel. Implanted hydrogel provides favorable environments for the survival and maturation of the newly generated neurons. Collectively, these results suggest a beneficial effect of biomaterial after a cortical traumatic injury.

## 1. Introduction

Traumatic brain injury (TBI) is defined as damage to the brain caused by an external mechanical force. TBI is a major cause of morbidity and mortality throughout the world. TBI in adult brain induces neural tissue loss, inflammation, and reactive astrogliosis [1,2], which contribute to secondary tissue loss, impaired regeneration, and associated functional disabilities. The inflammation following the lesion plays a very important role in preventing the onset of infections by isolating the impacted area and cleaning up cellular debris. These cells will also secrete multiple trophic factors in the extracellular environment, with pro-inflammatory effects to recruit other immune cells but also anti-inflammatory effects to reduce the release of toxic substances and support cell survival [3,4]. All these damages are responsible for long-term symptoms, such as deficits in motor function if the corresponding cortical area is affected. Despite cerebral plasticity, the human brain displays a poor ability to self-repair, and there are no curative treatment options [5,6].

Several emerging regenerative approaches have been developed to promote regeneration and functional recovery following injury. We have previously shown that the transplantation of neural progenitor cells (NPCs) obtained from fetal tissues [7,8,9] or differentiated from embryonic stem cells [10,11] has resulted in the partial restoration of damaged cortical pathways. In order to improve the therapeutic effectiveness of cell transplantation, combinatorial approaches are currently under investigation. We have also shown that a delay between the cortical lesion and cell transplantation can enhance graft vascularization, survival and projections associated with better functional recovery [9,12].

To further improve the therapeutic effectiveness of transplantation and functional recovery after injury, biomaterials protecting grafted cells and/or supporting repair processes, such as extracellular matrix (ECM) substitute, are currently in development. ECM constitutes above 20% of the total central nervous system composition [13]. Hyaluronic acid (HA) is considered the scaffold of neuronal ECM [14], and HA-based biomaterials are receiving increased attention in tissue engineering because of their unique and appealing biological properties such as biocompatibility, biodegradability, and nontoxicity. Before combining the cell with the HA hydrogel for transplantation, it is important to know the effects of the implanted HA hydrogel alone in the host brain. Here, we investigated the therapeutic effect of HA on host tissue after a cortical trauma. HA exists in different fragments lengths, each with specificities: high molecular weight (more than 3 × 10^6^ Da) has been shown to inhibit remyelination and oligodendrocytes maturation [15] but will have an anti-inflammatory effect [16]; intermediate molecular weight (between 2 × 10^4^ Da to 3 × 10^6^ Da); low molecular weight (LMW, less than 2 × 10^4^ Da), promotes angiogenesis but will have a pro-inflammatory effect [16]. In this study, we used a commercial kit of hyaluronan-based biomaterial with an intermediate length (above 300 kDa). In addition, the rigidity of this type of biomaterial in the form of a hydrogel (around 300 Pa) must be considered to fit with brain rigidity (between 100 and 500 Pa, [17]). As hyaluronan alone does not permit cell attachment [18,19], we added other ECM components, such as laminin and fibronectin [14], along with a synthetic amino acid called poly-L-ornithine [20].

In the present study, we analyzed the short-term (one week) and long-term (one month) effects of a HA implanted on host tissue immediately or with a delay after cortical traumatic injury. For this, we implanted biomaterial into the motor cortex of adult mice immediately after the lesion or 7 days later. Then, we analyzed its impacts on the secondary lesion, host cell migration, vascularization of the biomaterial and inflammation of host tissue. We have shown that implanted hydrogel provides a favorable environment for the survival and maturation of the newly generated neurons.

## 2. Materials and Methods

### 2.1. Hyaluronic Acid-Based Hydrogel Preparation

Hyaluronic acid-based hydrogel was prepared freshly according to the manufacturer’s protocol (HyStem^®^ Cell Culture Scaffolds, Sigma-Aldrich, St. Louis, MO, USA). Briefly, HyStem and Extralink^®^ stock solutions were reconstituted with degassed water and gently shacked until dissolution. In total, 50 µg/mL of poly-L-ornithine (Sigma-Aldrich, St. Louis, MO, USA), 5 µg/mL of laminin (Invitrogen, Waltham, MA, USA) and 5 µg/mL of fibronectin (Sigma-Aldrich, St. Louis, MO, USA) were added to HyStem to improve cell adhesion, before combination with Extralink^®^ in a 2:0.5 ratio. In the no-delay group, due to a lack of tissue cavitation immediately after TBI, HA hydrogel was injected into the center of cortical impact. In the delayed group, hydrogel was injected into the lesion cavity.

### 2.2. Animals

Animal housing and experimental procedures were carried out in accordance with the guidelines of the French Ministry of Agriculture and Food (decree 87849) and of the European Communities Council Directive (2010/63/EU). The procedures, referenced under the file number APAFIS#23295-2019121217062497 v2, were approved by ethics committee N°84 COMETHEA New-Aquitaine Region and authorized by the French Ministry of Higher Education, Research and Innovation. All efforts were made to reduce the number of animals used and their suffering. A total of 42 male and female adult (5 to 7 months old) C57BL/6 mice were used in this study: all mice were lesioned, among which 16 were used as the “lesion” group, 13 mice were implanted immediately after the lesion (“no delay” group) and 13 mice were implanted 7 days later (“delay” group; Figure 1).

### 2.3. Lesion and Implantation Procedure

Animals were anesthetized with an intraperitoneal injection of xylazine/ketamine solution (10 and 100 mg/kg, respectively), and the left motor cortex was lesioned with a controlled computer impact device (CCI, Pinpoint PCI3000 Precision Cortical Impactor, Hatteras Instruments, Cary, NC, USA) through a craniotomy (0.5 to 2.5 mm lateral to medium line and 0.5 to 2.5 mm rostral to Bregma) at a velocity of 4 m/s, 2 mm diameter and 1 mm depth [21]. Immediately after the lesion, 1.5 µL of hydrogel in liquid form was slowly injected with a 10 µL Hamilton syringe into the lesioned cortex of the “no delay” group. The syringe was maintained for a few seconds to avoid backward surge and to improve hydrogel diffusion into the tissue. For the “delay” group, implantation was carried out 7 days after the motor cortex lesion [9].

### 2.4. BrdU Injections

To determine newly generated cells, mice were given a single intraperitoneal injection with bromodeoxyuridine solution (BrdU, 50 mg/kg, NaOH 0.1 M, NaCl 0.9%, Sigma-Aldrich, St. Louis, MO, USA) the day of lesion or implantation and sacrificed at different time points.

### 2.5. Tissue Preparation and Immunohistochemistry

At different time points, mice were injected intraperitoneally with a sublethal dose of pentobarbital (911 mg/kg, Doléthal, Vetoquinol, Paris, France) and perfused transcardially with 150 mL of saline solution (0.9%) followed by 200 mL of ice-cold paraformaldehyde (PFA) 4% in phosphate buffer 0.1 M (PB, pH 7.4). Brains were removed and post-fixed in PFA 4% overnight at 4 °C. Brains were cut into 40 µm-thick coronal sections on a freezing microtome (Leica, Boston, MA, USA), in 6 series and stored in a cryoprotective solution (glucose 20%, ethylene glycol 40%, sodium azide 0.025% and phosphate buffer 0.05 M, pH 7.4) at −20 °C.

Free-floating sections were incubated in blocking solution (0.3% Triton X-100, 3% donkey serum in Tris-buffer solution (TBS) 0.1 M pH 7.6) for 90 min at room temperature (RT). Primary antibodies diluted in blocking solution were incubated overnight at 4 °C. Appropriate donkey-secondary antibodies conjugated with Alexa Fluor fluorochromes were diluted in blocking solution and incubated for 1 h at RT. The following antibodies were used for: vessels with rat anti-CD31 (1:250, BD Biosciences 553370, East Rutherford, NJ, USA), neurons with rabbit anti-βIII tubuline (1:100, Abcam 18207, Cambridge, UK), mature neurons with rabbit anti-neuronal nuclei (NeuN, 1:500, Abcam 177487, Cambridge, UK), neuroblasts with goat anti-doublecortin (DCX, 1:250, Santa Cruz Biotechnologies 8066, Dallas, TX, USA), oligodendrocytes with rabbit anti-Olig2 (1:500, Millipore AB9610, Burlington, MA, USA), astrocytes with rabbit anti-glial fibrillary acidic protein (GFAP, 1:1000, Abcam 4674, Cambridge, UK) and microglia with rabbit anti-Iba1 (1:500, Fujifilm Wako 019-19741, Osaka, Japan). Rat anti-CD86 (1:200, Abcam 119857, Cambridge, UK) and chicken anti-Arg1 (1:2000, Santa Cruz Biotechnologies 27430, Dallas, TX, USA) were, respectively, used for M1 and M2 phenotype. Rat anti-C3 (1:200, Abcam 11862, Cambridge, UK) and goat anti-mouse S100A10 (1:200, Abcam 187201, Cambridge, UK) were, respectively, used for A1 and A2 phenotype. For rat anti-BrdU (1:250, Bio-Rad AbD Serotec OBT0030, Luxembourg, Luxembourg) staining, before incubation with the blocking solution, sections were pre-treated with 2N HCl and 0.5% Triton X-100 in PBS for 30 min at 37 °C followed by 30 min incubation with Borax, pH 8.6 at RT. All sections were finally incubated with DAPI (1:10,000, Sigma, St. Louis, MO, USA) for nuclei staining and covered with DePeX mounting medium (VWR).

### 2.6. Data Acquisition and Quantification

For each mouse, images were acquired using Axio Imager M2 Apotome (Carl Zeiss, Macquarie Park, Australia) at ×20 magnification at the rostral, middle, and caudal part of the lesion or the biomaterial implant. These images were used with ZEN software (Carl Zeiss, Macquarie Park, Australia) to quantify positive cells immunostaining per surface (bring back per one square millimeter) or Mercator Pro software (Explora Nova, La Rochelle, France) to determine blood vessel density in the implant (total surface of blood vessels/total surface of the implant, in square millimeters). Areas of interest were further imaged using a laser-scanning confocal microscope (Evident Olympus FV3000, Nagano, Japan).

The implant volume was calculated with the following formula: V (mm^3^) = [(s1 + s2)/2] × d, where s = surfaces (in square millimeters) and d = distance (in millimeters) between two sections [9]. The tissue loss was calculated with the following formula: % Tissue loss = [(V_C_ − V_L_)/V_C_] × 100, where V_C_ is the volume of contralateral hemisphere and V_L_ is the non-injured tissue volume in the ipsilateral hemisphere subjected to TBI [22].

### 2.7. Statistical Analysis

Statistical analyses were performed using GraphPad Prism 9. Data are expressed as mean ± SEM and evaluated using one or two-way analysis of variance (ANOVA) followed by a Tukey correction or Mann–Whitney test. Differences were considered statistically significant when *p* < 0.05, *p* < 0.01, *p* < 0.001, *p* < 0.0001 (*, **, ***, ****; respectively).

## 3. Results

### 3.1. The Impact of the Hydrogel on Tissue Cavity Formation

To evaluate the impact of hydrogel on the tissue cavity formation and resulting cortical tissue loss, the percentage of cortical tissue loss was calculated one week and one month following cortical lesion and hydrogel implantation. One week after cortical injury, in the lesioned group, the necrotic tissue was not yet eliminated and only a small lesion cavity was observed (Figure 2A), and the quantitative analysis showed that the percentage of tissue loss was 25.25 ± 2.64% (Figure 2G). In the no-delay implanted group, the lesion cavity limited to the interface between the hydrogel and the host tissue (Figure 2C), the quantitative analysis showed that the percentage of cortical loss was 0.15 ± 4.52% (Figure 2G). In the no-delay implanted group, the percentage of tissue loss was smaller compared to that of the lesioned group and delay implanted hydrogel group (*p* < 0.0001). In the delay implanted group, quantitative analysis showed that the percentage of cortical loss was 28.85 ± 2.33% (Figure 2G). Implantation of the hydrogel without delay shows a greater preservation of cortical tissue compared with no delay implanted hydrogel. We next investigated the evolution of the lesion cavity volume between one week and one-month post-surgery. In the injured group, the percentage of cortical loss was 33.39 ± 4.24% (Figure 2B). Interestingly, the percentage of cortical loss decreased in the no-delay hydrogel implanted group (20.46 ± 1.67%, *p* = 0.0022). In the delayed hydrogel implanted group, quantitative analysis showed that the % of cortical loss was 30.29 ± 2.03% (Figure 2G). These data suggest that the implantation of hydrogel without delay efficiently prevented the progression of the lesion in the short term.

### 3.2. The Impact of the Hydrogel on Angiogenesis

Angiogenesis has a major role in the repair of brain damage. To assess the angiogenic effect of hydrogel, immunohistochemical staining of CD31 was performed after one week or one month of hydrogel implantation. For this, we analyzed the vascular density in the implanted hydrogel by quantifying the area of blood vessels labeled with CD31 relative to the total area of the hydrogel. We found that the hydrogel was highly vascularized by the host as early as one week after implantation, whatever the time of its implantation (no delay: 2.1 ± 2.0 × 10^5^ mm^2^; delay: 2.1 ± 2.0 × 10^5^ mm^2^; Figure 3A,C,E). At one-month post-implantation, the hydrogel CD31 density increased, and this increase was much more pronounced for the no-delay implanted hydrogel compared to the delay implanted hydrogel (no delay: 3.3 ± 2.7 × 10^5^ mm^2^; delay: 2.8 ± 2.1 × 10^5^ mm^2^; *p* = 0.015; Figure 3B,D,E). These results suggest that the presence of a delay between the time of injury and hydrogel implantation does not impact the vascularization of the hydrogel.

### 3.3. SVZ Host Neuroblasts Migration to the Implanted Hydrogel

Neuroblasts generated in the adult subventricular zone (SVZ) migrate via the rostral migratory stream (RMS) to the olfactory bulb (OB) where they differentiate into the OB interneurons. We have previously shown that the cortical lesion stimulates the production of adult SVZ neuroblast as well as the ectopic migration of a part of these neuroblasts to the site of cortical injury [23]. Here, we investigated the impact of hydrogel implantation into the cortical lesion on host neuroblast cell migration. For this, we quantified the number of doublecortin (DCX) positive cells, a microtubule-associated protein expressed by migrating neuroblasts in the cortex as well as in the hydrogel implanted immediately or one week after cortical lesion.

One week after the cortical lesion or hydrogel implantation, DCX+ cells were already present in the cortex adjacent to the injured cortex in all groups (Figure 4A–C,A′–C′). Quantification of the number of DCX+ neuroblasts in the cortex at one week, showed more DCX+ cells in the cortex of the injured group compared to both hydrogel implanted groups (Figure 4D; two-way ANOVA). Analysis of the neuroblast migration, one month after injury or hydrogel implantation, also showed more DCX+ cells in the cortex of the injured group compared to hydrogel implanted groups (Figure 4D). We next evaluated the presence of DCX+ cells within hydrogel (Figure 5A–D). The analysis of the results showed a similar number of DCX+ cells in the hydrogel implanted with a delay compared to no-delay implanted group, both at one week and one month after implantation (Figure 5E,F; Mann–Whitney test). The presence of DCX+ cells in the hydrogel may explain the fact that we observed fewer DCX+ cells in the cortex of the implanted groups compared to the injured group.

Next, we investigated the factors that support the migration of DCX+ cells from the SVZ to the cortex. Previously, we have shown ectopic migration of DCX+ neuroblasts from the SVZ to the site of cortical injury and that the majority of DCX+ cells migrate to the cortex along with blood vessels or glial cells [23]. To determine whether the presence of hydrogel impacts this mode of migration, we performed a triple immuno-labeling combining DCX, CD31, GFAP and quantified the number of DCX+ cells either in association with blood vessels or with astrocytes one week and one month after the lesion or hydrogel implantation. In the cortex of lesioned group, one week after the lesion, we found 41 ± 8% of DCX+ cells migrating in association with blood vessels (Figure 6A) and 24 ± 7% of DCX+ cells in association with astrocytes (Figure 6D). In the cortex of the hydrogel implanted groups, one week after the implantation, we found 38 ± 7% and 36 ± 6% of DCX+ cells in association with blood vessels, respectively, in no-delay (Figure 6B) and in delay groups (Figure 6C). Concerning the migration of neuroblasts in association with the astrocytes, we found, in the cortex of the hydrogel implanted groups, 32 ± 9% and 30 ± 10% of DCX+ cells in association with astrocytes, respectively, in the no-delay group (Figure 6E) and in the delay group (Figure 6F). We also analyzed the results one month after the lesion and or implantation. We did not see statistically significant differences between different groups (two-way ANOVA) at one week or one month after the lesion and or implantation, suggesting that the implanted hydrogel does not impact the mode of migration of neuroblasts to the cortex following a cortical injury.

### 3.4. Implanted Hydrogel Provides Structural Support to the Host Cells

Host cells, evidenced by nuclear DAPI labeling, were infiltrated into the hydrogel in both groups of the implanted hydrogels. We performed fluorescent immunohistochemical analysis to determine the identity of the host cells infiltrated into the hydrogel. In addition, to determine whether the cells that have migrated into the hydrogel are newly generated cells or cells already present in the host, mice were injected with BrdU the day of lesion or implantation (Figure 1).

We found a large number of Olig2+ cells in both hydrogel implanted groups, and the number of oligodendrocytes was similar within hydrogels at one week and one month (Figure 7). Indeed, the number of oligodendrocytes/mm^2^ for the no-delay group was 787 ± 110 at one week and 2109 ± 329 at one month (Figure 7A,B,E). The number of oligodendrocytes/mm^2^ for delay implanted hydrogel was 971 ± 94 at one week and 1586 ± 1866 at one month (Figure 7C–E). Among these oligodendrocyte cell populations, at one month, we found 296.7 ± 125.8 Olig2+ /BrdU+ cells per mm^2^ in the no-delay implanted hydrogel group (Figure 8A–D), and 395.8 ± 195.3 Olig2+ /BrdU+ cells per mm^2^ for the delay implanted hydrogel group (Figure 8D–G).

The second population of the host infiltrated cells to the hydrogel was GFAP+ cells. Quantification of the number of GFAP+ cells within hydrogel revealed that the number of astrocytes in the delay implanted group was similar to that of the no-delay group (7492 ± 507 no delay and 6988 ± 1155 delay; one-way ANOVA). Among GFAP+ cells, 554 ± 119 were Brdu+ (7%) in the no-delay implanted group and 1038 ± 128 were BrdU+ (14%) in the delay group (Figure 8H–N), indicating that they are newly generated astrocytes.

Next, we assessed the presence of immature and mature neurons within the hydrogel using, respectively, DCX and NeuN markers. First, the number of DCX+ cells within the hydrogel was similar in the delay and no-delay hydrogel implanted groups (Figure 5). Second, the number DCX+ cell in the hydrogels was slightly higher, one week after hydrogel implantation compared to that of one month (Figure 5). We also quantified, one-month post-implantation, the number of DCX+/BrdU+ cells and found that a small fraction (no delay: 13%; delay: 11%) of the cells was double labeled (Figure 9), suggesting that they are newly generated neuroblasts.

We also investigated the presence of mature neurons within the hydrogel. Interestingly, we found a large number of βIII-tubulin+ and NeuN+ neurons, within hydrogel, in the no-delay (Figure 10A,B,E,F) and delay (Figure 10C–F) hydrogel implanted groups. It is important to know if the neurons present in hydrogels are newly generated neurons or neurons already present in the host that migrated to the hydrogel. For this purpose, we analyzed the mature neurons containing BrdU and found numerous NeuN+/BrdU+ within the hydrogel in both conditions, indicating that they are newly generated neurons (Figure 9H–N). Next, we quantified the number of NeuN+/BrdU+ in hydrogels and found more NeuN+/BrdU+ in hydrogel implanted with delay, but the difference was not statistically significant (Mann–Whitney test; Figure 9K).

### 3.5. Effects of Implanted Hydrogel on Glial Scar Formation, Microglia Activation and Neuroinflammation

Traumatic brain injury induces astrocyte reactivity as well as glial scar formation, microglia activation and neuroinflammation, these alterations may impair axonal regeneration [24,25,26]. Thus, approaches are needed to make the environment more permissive to axonal regeneration. We evaluated the effects of implanted hydrogel on glial scar formation, microglia activation and neuroinflammation.

We analyzed the thickness of the glial scar and the number of GFAP+ cells at different distances (0 to 100, 100 to 200 and 200 to 300 µm) of the cortical lesion in the presence or absence of the hydrogel. Glial scars were recognized by intense GFAP immunostaining around the cortical lesion cavity. One month after the surgery, we found that the thickness of the glial scar was relatively smaller in the hydrogel implanted with the delay group (lesion: 34 ± 3.9 µm; no delay: 26 ± 5.3 µm; delay: 21 ± 1 µm; Figure 11A–D). The number of GFAP+ cells also tended to decrease in the hydrogel implanted with delay group compared to the lesioned group (One-way ANOVA, *p* = 0.069). Indeed, in the first 100 µm from the lesion border, the number of GFAP+ cells per mm^2^ was 12,515 ± 1347 in the lesioned group, 9178 ± 1170 GFAP+ in the no-delay hydrogel implanted group and 8142 ± 594 in the delay implanted group. We did not observe any difference among the different groups in the number of GFAP+ cells beyond 100 µm from the lesion cavity or from the implanted hydrogel.

In addition to the change in number, the reactive astrocytes show a morphological change. Indeed, in the lesioned group, reactive astrocytes display enlarged soma with short processes (Figure 11A,A′), which is not the case for the astrocytes found in the hydrogel implanted groups, where they adopted elongated and branched processes (Figure 11B,B′,C,C′).

Microglia activation, the primary mediator of inflammation, is a major component of gliosis and neuronal loss following cortical trauma. We counted the total number of microglial cells in the cortex ipsilateral to the lesion or implantation. The number of microglial cells in the cortex was 10,699 ± 1926 in the lesioned group, 9431 ± 2257 in the no-delay group and 9181 ± 1752 in the delay group. At one-month post-injury, in the lesioned group, many of the microglial cells exhibited an amoeboid morphology in the cortex, around the lesion cavity. In the cortex of the hydrogel implanted groups, most of the microglial cells exhibited a ramified morphology.

### 3.6. Astrocytes and Microglia/Macrophage Polarization after Hydrogel Implantation

Astrocytes and microglial cells can induce both protective and toxic actions, which can impact the extent of brain damage or its repair after injury. In this context, we asked whether the presence of the implanted hydrogel impacts the polarization of the astrocytes. In order to examine the pro-inflammatory vs. anti-inflammatory profile of astrocytes and microglia, immunohistochemistry was used to identify astrocytes and microglia/macrophage subpopulation in the cortical areas adjacent to the lesion and hydrogel implantation. For this, double immunofluorescence was used to specifically identify GFAP+ or Iba1+ cells with a pro- or anti-inflammatory phenotype. Concerning GFAP+ cells, C3 was used as a marker for the pro-inflammatory phenotype, whereas S100A10 was used for the anti-inflammatory phenotype. For Iba1+ cells, CD86 was used as a marker for the pro-inflammatory phenotype, whereas Arginase-1 (Arg1) was used for the anti-inflammatory phenotype.

One month after the lesion, 11.8 ± 1.7% of the astrocytes present around the lesion colocalized with the pro-inflammatory marker C3 (Figure 12D,G), while only 1.5 ± 1.9% showed an anti-inflammatory phenotype by colocalizing with the marker S100A10 (Figure 12A,G).

Interestingly, in the no-delay hydrogel implanted group, in the cortex around the lesion/implantation, the number of pro-inflammatory astrocytes decreased to 5.8 ± 1.8% (Figure 12D,G), while the number of anti-inflammatory astrocytes increased to 5.87 ± 1.96% (Figure 12A,G). In the delay hydrogel implanted group, the number of pro-inflammatory astrocytes also decreased to 6.3 ± 0.6% (Figure 12E,G). However, the number of anti-inflammatory astrocytes increased only slightly to 1.9 ± 1.0% (Figure 12B,G) compared to the lesioned group.

Concerning the microglia polarization, we observed that 2.0 ± 0.9% of the microglia present around the lesion colocalize with the pro-inflammatory marker CD86 while 2.8 ± 1% colocalize with the anti-inflammatory marker Arg1 (Figure 13A,D,G). Thus, the balance between the pro- and anti-inflammatory phenotype is at equilibrium in the cortex adjacent to the lesion one month after the lesion. In the no-delay hydrogel implanted group, 2.9 ± 0.9% of microglia colocalized with the pro-inflammatory marker CD86 and 3.6 ± 0.7% of microglia colocalized with the anti-inflammatory subtype (Figure 13B,E,G). In the delay hydrogel implanted group, 1.8 ± 0.6% of microglia colocalized with the pro-inflammatory marker CD86 and 5.3 ± 0.3% of microglial cells present in the cortex colocalized with the anti-inflammatory marker Arg1 (Figure 13C,F,G). Implantation of the hydrogel with a delay would be more favorable than implantation immediately after injury to modulate the polarization state of microglial cells and promote beneficial anti-inflammatory effects.

## 4. Discussion

Traumatic brain injury results in massive neuronal death as a result of both direct and secondary injury. Hyaluronan-based biomaterials applied to damaged tissue create appropriate support and a microenvironment favorable for tissue repair. Here, we investigated the therapeutic effect of HA hydrogel on host cortical tissue after a TBI. We observed that implanted hydrogel limits the progression of cortical injury, decrease the glial scar formation, and provide favorable environments for the survival and maturation of the newly generated neurons. Collectively these results suggest a beneficial effect of biomaterial after a traumatic cortical injury.

Biomaterial vascularization is an essential factor for the survival of endogenous or exogenous cells after transplantation. It has been reported that hyaluronan hydrogels have an adequate porosity to be vascularized [27,28]. We observed the vascularization of the implanted HA as soon as one-week post-implantation. The proangiogenic effects of HA have been reported in several studies. Interactions between endothelial cells and ECM components are crucial in the formation of a new vessel. CD44 and HA-mediated motility (RHAMM), two main receptors of HA, are present on the surface of the endothelial cells. The implanted HA in our model may stimulate HA receptors present at the surface of the host endothelial cells and stimulate angiogenesis. It has been shown that LMW HA stimulates vascular endothelial cell proliferation, migration, and vessel formation [29,30]. This rapid vascularization of the implanted hydrogel demonstrates the great potential of HA to promote vascularization in tissue engineering.

Another highlight of our study is the presence of newly generated neurons in the hydrogel. We have previously shown that motor cortical lesion stimulates the migration of SVZ-generated neuroblasts to the site of injury [23]. However, the survival of these migrating neuroblasts is very limited, and the majority of these neuroblasts die and do not integrate into the host circuitry. Interestingly, in the present study, we observed the clusters of newly generated immature and mature neurons within the implanted hydrogel. The rapid vascularization of the hydrogel may participate in the survival and maturation of neurons within the hydrogel. One of the important questions to elucidate is the functionality of these neurons.

The infiltration of cells from the host brain tissue into the hydrogel may be impacted by the formation of a glial scar after the lesion. Previous studies have shown that the hyaluronan-based biomaterial induced glial scar reduction in tissue removal models of cortical lesions [28,31], as well as in stroke models [32,33]. In accordance with these studies, we observed that the implantation HA induced a reduction in the extent of glial scar formation. The decrease in astrocyte activation [34] and glial scar formation would occur thanks to HMW hyaluronan through the decrease in chondroitin sulfate proteoglycans (CSPG) expression, as shown in primary astrocytes cultures and in a spinal cord injury model [35]. The reduction in glial scarring observed after HA implantation may ultimately provide a more favorable post-injury environment for tissue repair.

Finally, one of the desired effects of biomaterials is their ability to modulate injury-induced neuroinflammation. We observed a potential reduction in pro-inflammatory astrocytes in the cortex of no-delay and delay HA implanted groups. We found an increase in anti-inflammatory astrocytes only in the delay HA implanted group. These results show that the presence of the implanted hydrogel impacts the polarization of the astrocytes in favor of a neuroprotective effect. Regarding the modulation of microglia, we observed a potential increase in the anti-inflammatory microglia subtype only when the hydrogel was implanted with a delay. Implantation of the hydrogel with a delay would be more favorable than implantation immediately after injury to modulate the polarization state of microglial cells and promote beneficial anti-inflammatory effects. It has been reported that the LMW hyaluronan activates macrophages and microglia to a pro-inflammatory subtype, while HMW hyaluronan promotes an alternative activation to an anti-inflammatory subtype [36].

For clinical application, the timing of hydrogel implantation is crucial. Our results concerning the presence or absence of a delay between the cortical lesion and hydrogel implantation and the effects of the hydrogel observed are variable and depend on the phenomenon studied. For instance, we have shown that immediate hydrogel implantation after TBI may be more effective than one week delaying hydrogel implantation as a decrease in tissue loss or an increase in vascularization. These results can be explained by the fact that one week after the injury, a part of the tissue loss has already occurred, and implantation of hydrogel with a delay does not prevent the progression of the lesion. Regarding the vascularization, it is not surprising that the vascular density is more important in the immediately implanted hydrogel; indeed, the fact that the hydrogel is in place earlier allows for faster vascularization of the hydrogel, and thus, a greater density of vessels is observed.

Implantation with a delay has a beneficial effect on the relatively later processes, such as modulation of neuroinflammation. Indeed, we observed that the implantation of the hydrogel with a delay would be more favorable to modulating the polarization state of microglial cells and promoting beneficial anti-inflammatory effects.

A possible limitation of this study is the lack of a functional assessment of hydrogel-implanted animals to verify if the presence of hydrogel can modify the motor deficits caused by the motor cortex lesion. However, we believe that the changes induced by the implantation of the hydrogel alone may not be sufficient to induce motor recovery and that it is necessary to combine the hydrogel with cell therapy to improve the efficiency of the transplantation. We have previously shown that the transplantation of neural progenitor cells obtained from fetal tissues [7,8,9] or differentiated from embryonic stem cells [10,11], has resulted in the partial restoration of damaged cortical pathways. We have also shown that a delay of one week between the cortical lesion and cell transplantation can enhance graft vascularization, survival and projections associated with better functional recovery [9,12]. To further improve the regeneration capacity of damaged cortical circuitries and facilitate the recovery of lost function, a combination of approaches, such as the combination of hydrogel and cells should be considered in TBI treatment.

One of the major challenges facing cell-based therapies is the survival and engraftment of transplanted cells following implantation into the host tissue. One possible application of HA-based hydrogel is their combination with cells intended for transplantation to improve the cell survival of the graft. In line with the rapid vascularization of the implanted hydrogel that we observed, we expect that the combination of cells with HA hydrogel will allow a better vascularization of the graft, which should improve the survival of the transplanted neurons and, consequently, projections developed by the grafted neurons.

In summary, the implantation of a hyaluronan-based hydrogel may ameliorate tissue repair following TBI. The rapid vascularization of the hydrogel, the reduction in glial scarring and the evidence of a pro- to anti-inflammatory activity collectively suggest a beneficial effect of biomaterial after a traumatic cortical injury.

## Figures and Tables

**Figure 1 cells-11-03831-f001:**
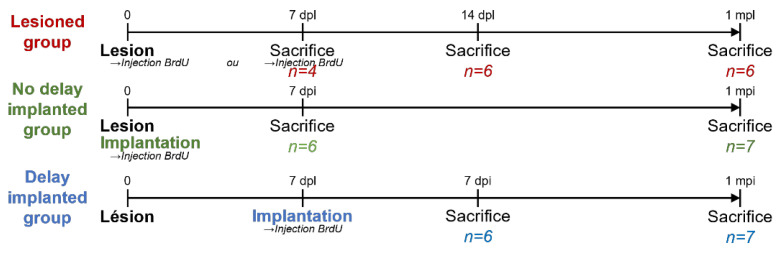
Timeline of the study representing the different groups and the different time points of mice hydrogel implantation and sacrifice. dpl: days post-lesion; mpl: month post-lesion, dpi: days post-implantation, mpi: month post-implantation.

**Figure 2 cells-11-03831-f002:**
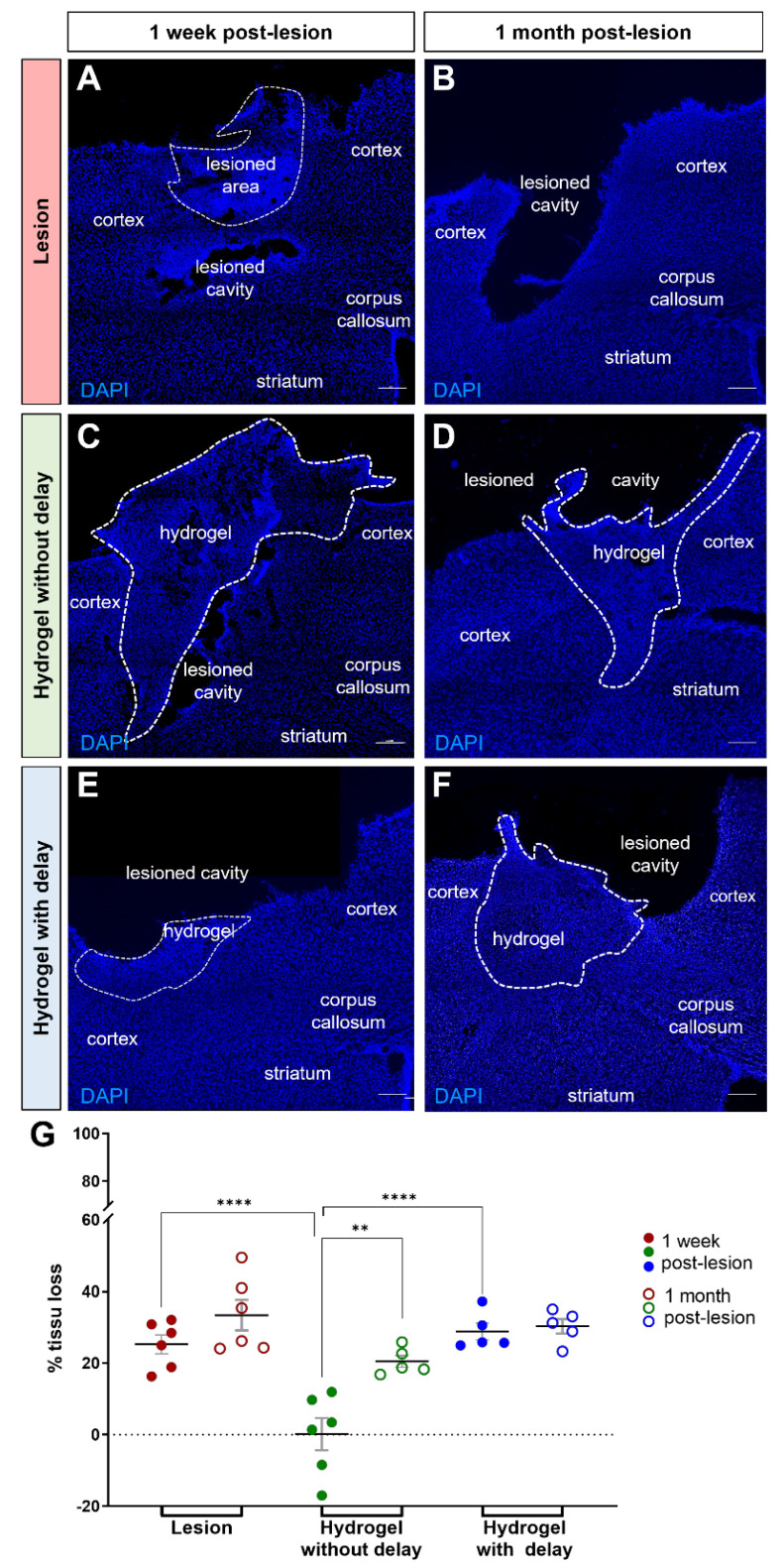
Cortical tissue loss one week and one month after the lesion. (**A**–**F**) Representative images of cells labeled with DAPI (blue) in lesioned (**A**,**B**) or implanted mice without delay (**C**,**D**) or with delay (**E**,**F**). Dashed lines indicate hydrogel localization. Scale bar: 200 μm. (**G**) Quantitative analysis of the cortical tissue loss over time in lesioned (red) or implanted mice without delay (green) or with delay (blue). Two-way ANOVA ** *p* < 0.01 **** *p* < 0.0001.

**Figure 3 cells-11-03831-f003:**
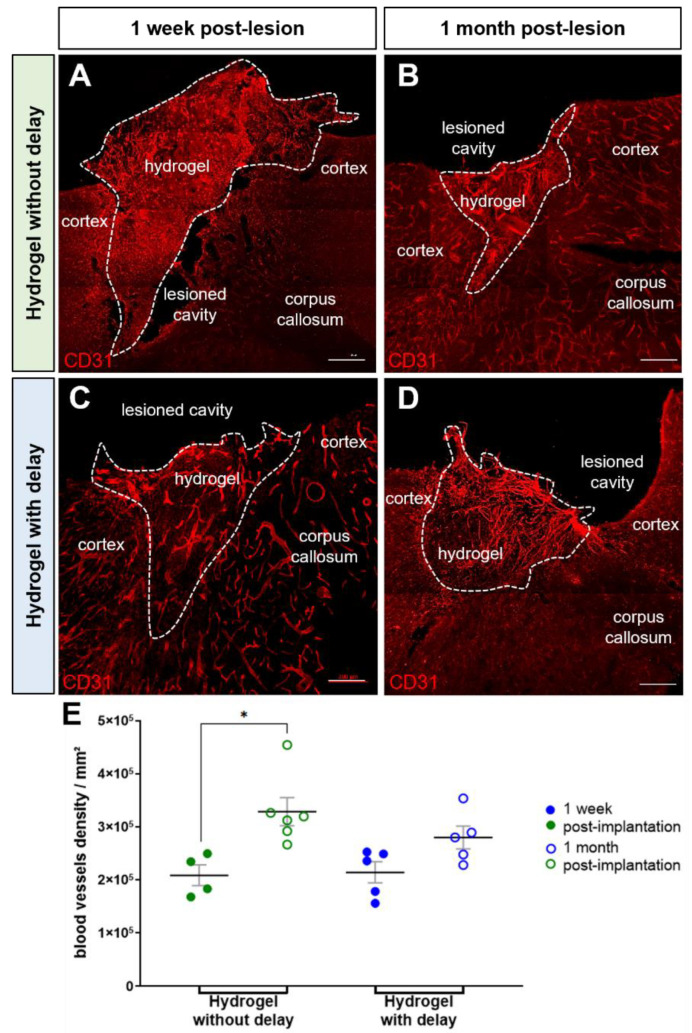
Hydrogel vascularization. (**A**–**D**) Representative immunofluorescence staining of blood vessels labeled with CD31 (red) in implanted mice in the no-delay group (**A**,**B**) or in delay group (**C**,**D**). Dashed lines indicate hydrogel localization. Scale bar: 200 μm. (**E**) Quantitative analysis of the blood vessels density over time in implanted mice in no-delay (green) or delay groups (blue). Two-way ANOVA * *p* < 0.05.

**Figure 4 cells-11-03831-f004:**
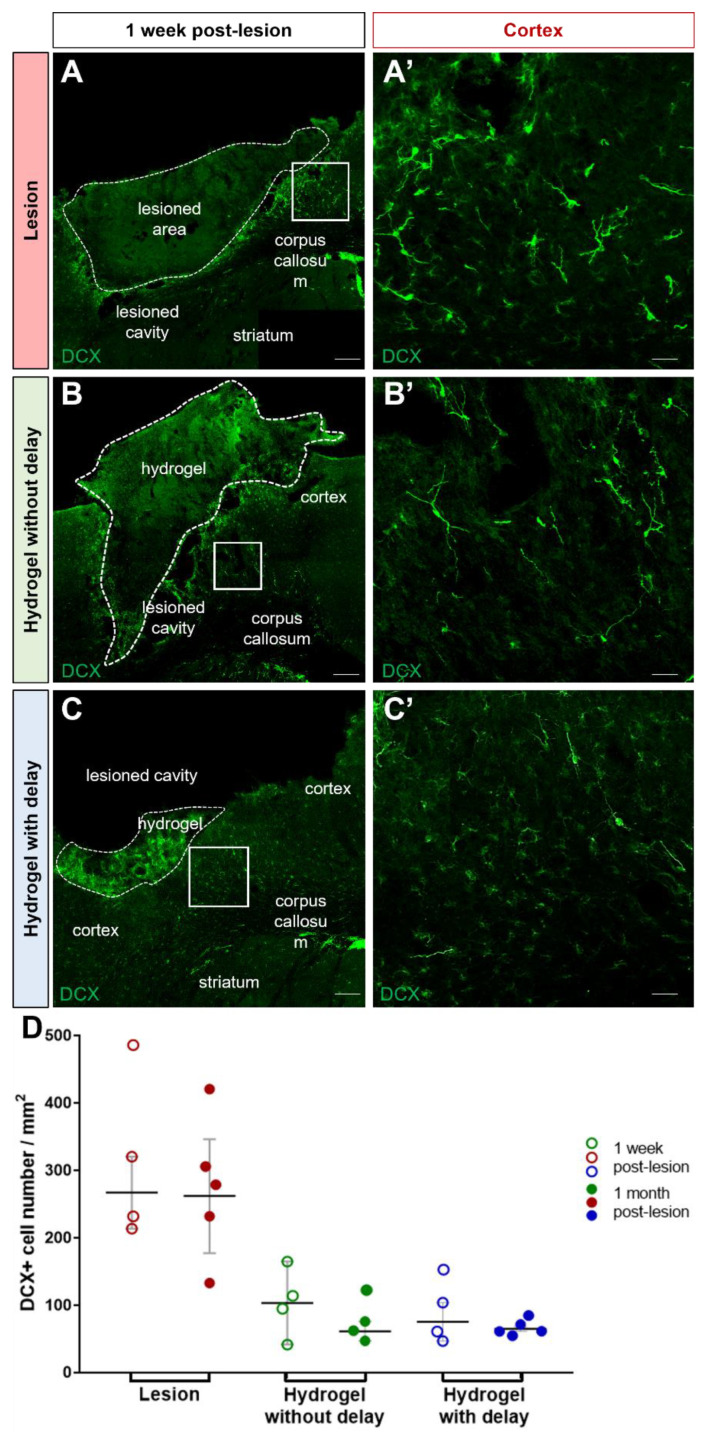
Neuroblasts migration to the cortex. (**A**–**C**) Representative immunofluorescence staining of neuroblasts labeled with DCX (green) in lesioned (**A**) or no-delay implanted (**B**) or delay implanted groups (**C**). Dashed lines indicate hydrogel localization. The squares indicate the area of magnification. Scale bar: 200 μm. (**A′**–**C′**) High magnification images of the neuroblast in the cortex. Scale bar: 20 μm. (**D**) Quantitative analysis of the number of the neuroblast over time in lesioned (red) or implanted mice without delay (green) or with delay (blue). Two-way ANOVA.

**Figure 5 cells-11-03831-f005:**
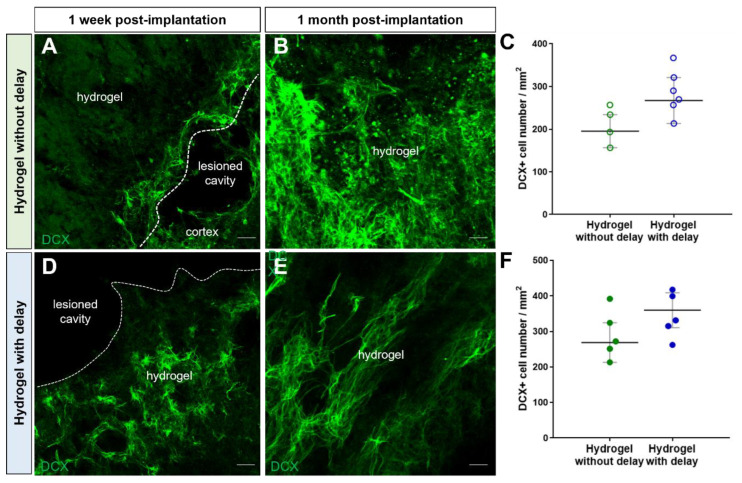
Neuroblasts migration into hydrogel. (**A**–**C**) Representative immunofluorescence staining of neuroblasts labeled with DCX (green) in delay (**A**,**B**) or no-delay implanted groups (**C**,**D**). Dashed lines indicate hydrogel localization. The squares indicate the area of magnification. Scale bar: 20 μm. (**E**,**F**) Quantitative analysis of the neuroblast number, one week (**E**) or one month (**F**) after implantation, in implanted no-delay (green) or delay implanted groups (blue). Mann–Whitney test.

**Figure 6 cells-11-03831-f006:**
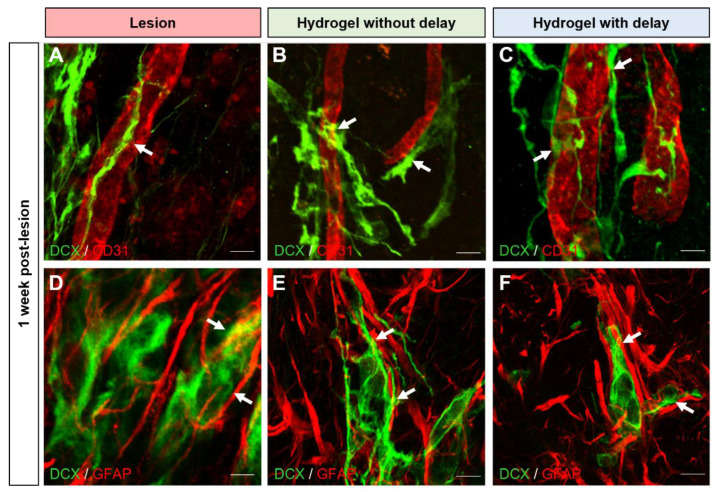
Mode of neuroblast migration towards cortical lesion site. (**A**–**F**) Representative immunofluorescence staining of neuroblasts labeled with DCX (green) and blood vessels with CD31 ((**A**–**C**), red) or astrocytes with GFAP ((**D**–**F**), red), in lesioned (**A**,**D**) or implanted mice without delay (**B**,**E**) or with delay (**C**,**F**). Dashed lines indicate hydrogel localization. The squares indicate the area of magnification. Scale bar: 20 μm.

**Figure 7 cells-11-03831-f007:**
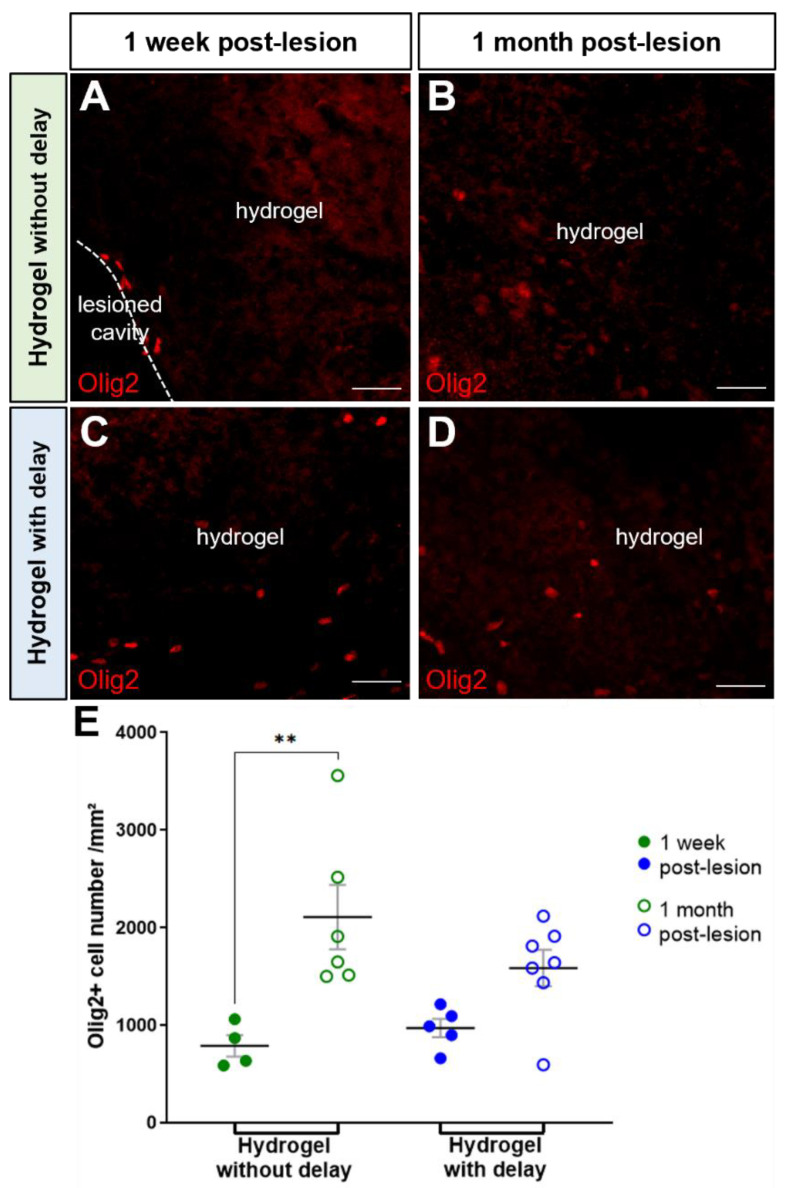
Oligodendrocytes into hydrogel. (**A**–**D**) Representative immunofluorescence staining of oligodendrocytes labeled with Olig2 (red) in no-delay (**A**,**B**) or delay implanted groups (**C**,**D**). Scale bar: 20 μm. (**E**) Quantitative analysis of the oligodendrocyte number into the corpus callosum over time in lesioned (red) or implanted hydrogel in no-delay (green) or delay groups (blue). Two-way ANOVA ** *p* < 0.01.

**Figure 8 cells-11-03831-f008:**
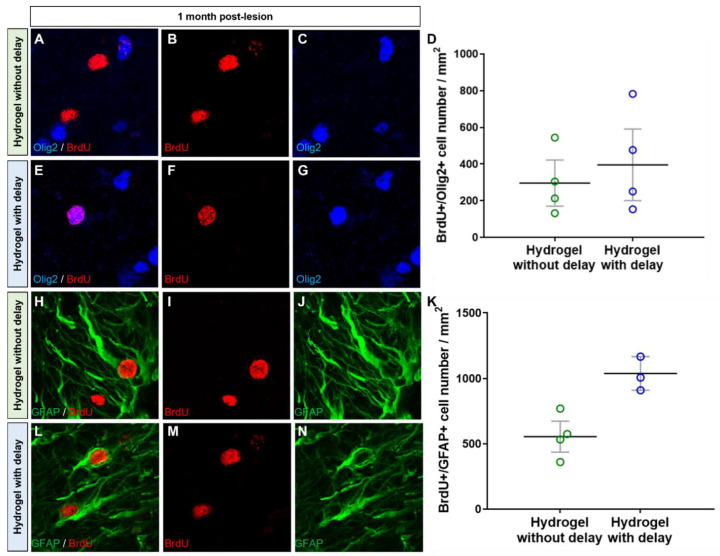
Newly generated oligodendrocytes and astrocytes into hydrogel. (**A**–**N**) Representative immunofluorescence staining of newly generated BrdU cells (red) colocalized with Olig2 (blue, (**A**–**G**)) or with GFAP (green, (**H**–**N**)) in delay (**A**–**C**,**H**–**J**) or no delay hydrogel implanted groups (**E**–**G**,**L**–**N**). (**D**,**K**) Quantitative analysis of the BrdU+/Olig2+ cells (**D**) or BrdU+/GFAP+ cells (**K**) in no-delay (green) or delay (blue) hydrogel implanted groups. Mann–Whitney test.

**Figure 9 cells-11-03831-f009:**
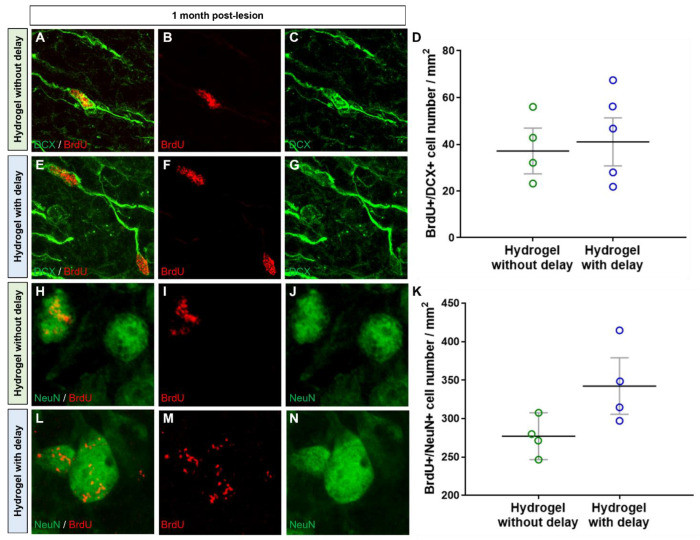
Newly generated neuroblasts and mature neurons into hydrogel. (**A**–**N**) Representative immunofluorescence images of newly generated BrdU+ cells (red) colocalized with DCX (green, (**A**–**G**)) or with NeuN (green, (**H**–**N**)) in hydrogel in no delay (**A**–**C**,**H**–**J**) or in delay (**E**–**G**,**L**–**N**) hydrogel implanted groups. (**D**,**K**) Quantitative analysis of the BrdU+/DCX+ neuroblasts (**D**) or BrdU+/NeuN+ neurons (**K**) within hydrogel delay (green) or no-delay (blue) groups. Mann–Whitney test.

**Figure 10 cells-11-03831-f010:**
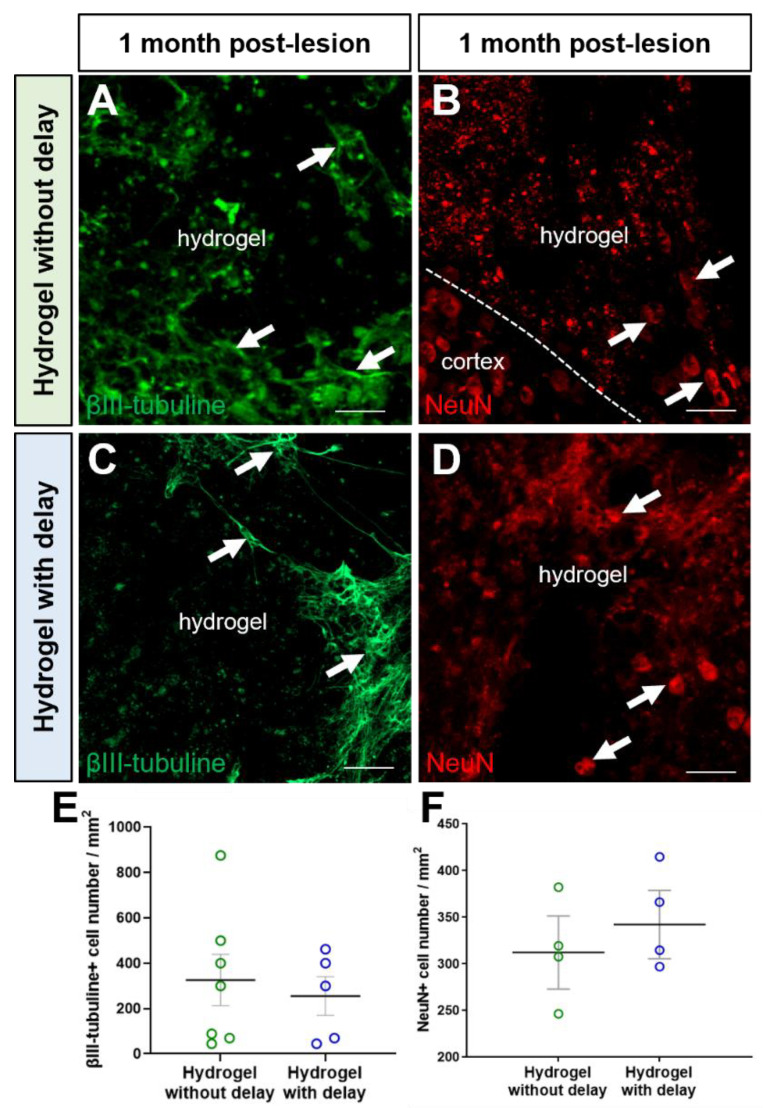
Neurons into hydrogel (**A**–**D**). Representative immunofluorescence staining of neurons labeled with βIII-tubulin (green, arrows, (**A**,**C**)) or with NeuN (red, arrows, (**B**,**D**)) in no-delay (**A**,**B**) or delay hydrogel implanted groups (**C**,**D**). Scale bar: 20 μm. (**E**,**F**) Quantitative analysis of the number of βIII-tubulin neurons (**E**) or NeuN+ neurons (**F**) in no-delay (green) or delay (blue) hydrogel implanted groups. One-way ANOVA.

**Figure 11 cells-11-03831-f011:**
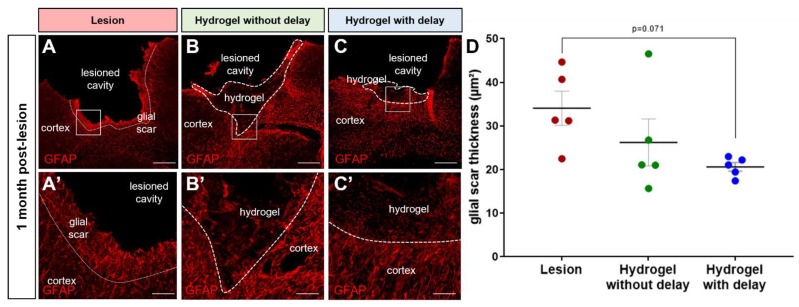
Glial scar thickness one month after the lesion/implantation. (**A**–**C**) Representative immunofluorescence images of GFAP+ cells (red) in lesioned (**A**) or delay (**B**) or no-delay (**C**) hydrogel implanted groups. Dashed lines indicate the limit of the glial scar (**A**) or hydrogel localization (**B**,**C**). The squares indicate the area of magnification. Scale bar: 200 μm. (**A′**–**C′**) High magnification images of the glial scar. Scale bar: 20 μm. (**D**) Quantitative analysis of the glial scar thickness in lesioned (red) in delay (green) or in no-delay (blue) hydrogel implanted groups. Groups were compared using one-way ANOVA, *p* = 0.07.

**Figure 12 cells-11-03831-f012:**
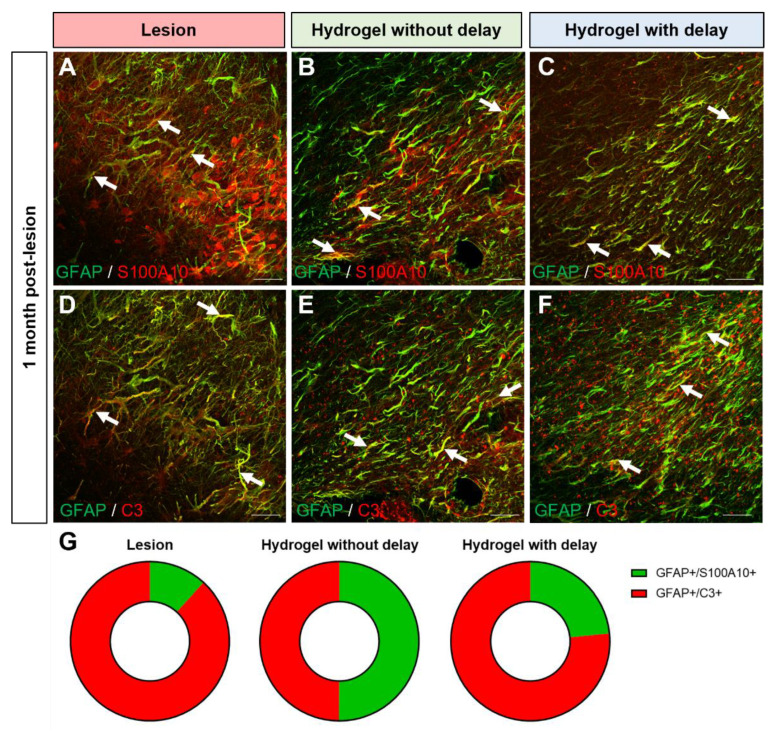
Anti and pro-inflammatory astrocytes in the cortex one month after the lesion/implantation. (**A**–**F**) Representative immunofluorescence staining of GFAP (green) colocalized with S100A10 ((**A**–**C**), red) or C3 ((**E**,**F**), red) in lesioned (**A**,**D**) in no-delay (**B**,**E**) or in delay (**C**,**F**) hydrogel implanted groups. Arrows indicate colocalization. Scale bar: 20 μm. (**G**) Representation of pro (red) and anti (green) -inflammatory astrocytes subtype proportion per condition. One-way ANOVA per subtype.

**Figure 13 cells-11-03831-f013:**
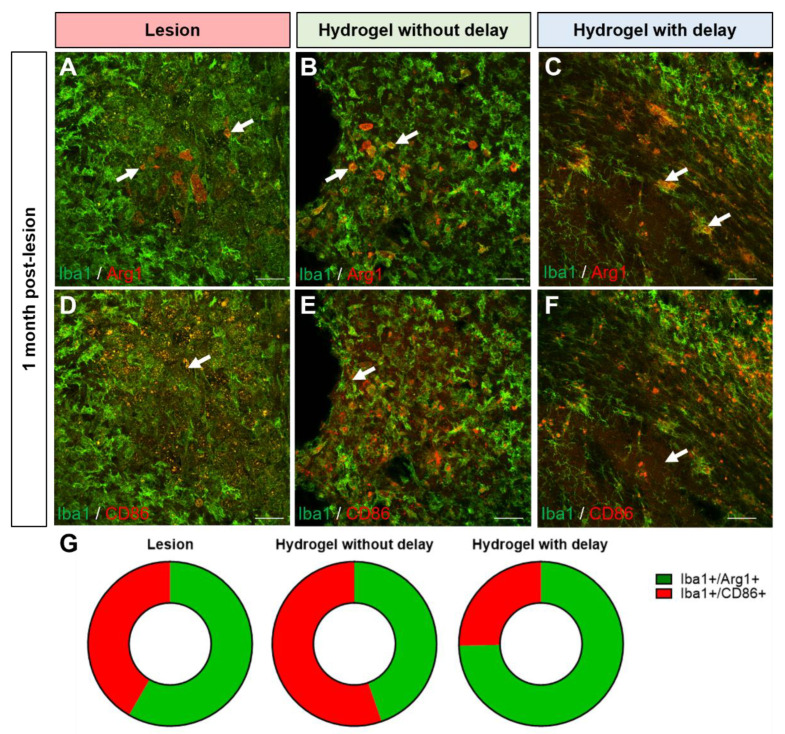
Anti and pro-inflammatory microglial cells in the cortex lesion one month after the lesion/implantation. (**A**–**F**) Representative immunofluorescence staining of Iba1+ cells (green) colocalized with Arg1 ((**A**–**C**), red) or CD86 ((**D**–**F**), red) in lesioned (**A**,**D**) or in no-delay (**B**,**E**) or delay (**C**,**F**) hydrogel implanted groups. Arrows indicate colocalization. Scale bar: 20 μm. (**G**) Representation of pro (red) and anti (green)-inflammatory microglia subtype proportion per condition. One-way ANOVA per subtype.

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
