# Peer review of "Beneficial Effects of Hyaluronan-Based Hydrogel Implantation after Cortical Traumatic Injury"

_cells, 2022, doi:10.3390/cells11233831_

Round 1

Reviewer 1 Report

This study is a histological study examining lesion, host cell migration and vascularization of the biomaterial and host tissue after hyaluronan-based hydrogels implantation in TBI. This reviewer has some major concerns with the study as it stands.

Major concerns

The emphasis of the study is immediate versus delay of transplantation but this is not apparent initially- please modify the title to reflect this and provide a rational in the introduction.

The authors do not adhere to the ARRIVE guidelines. This is a major issue for internal validity of the study.

Please state the sex of the mice used in methods and justify the choice.

Please state age of mice.

Please justify the unequal sample sizes (n=17 “control lesion”; n=12 “no delay” group and n=9 “delay” group) – the delay group is almost half of the control group (please see doi: 10.1111/bph.14442). In addition these n numbers don’t equal those in Fig1 G (n=11 no delay and n=10 delay). Please clarify. Please also calrify why the n numbers for the other figures are different- why and how animals were included/excluded. This is a concern given there was no randomisation or blinding reported.

Please clarify the staining for figure 1. This reviewer cannot distinguish the hydrogel from non-hydrogel tissue and the damaged from the non-damaged tissue for most of the figures. Please improve the images to improve interpretation and clarify how the hydrogel localisation was identified in the figures.

The values in the text (line 177) are not represented in figure 1G – ‘In delayed hydrogel implanted group, the size of lesion cavity is smaller than in other experimental group (p<0,0001)’- this is not the case in Fig1G as the no delay group is smallest. Please clarify. In addition the tissue loss in some of the animals is negative- please explain. Also why is tissue loss virtually zero in the no delay 1 week post-lesion group. This is particularly important as there is no randomisation or blinding reported. This is also important as this is the only group that has significant difference. This reviewer fails to see how the authors conclude ‘These data suggest that the implantation of hydrogel efficiently prevented the progression of the lesion.’ (line 187). Please review/clarify/modify accordingly.

For figure 2 there no lesion only group? This is important to evaluate the effect of the hydrogel.

Line 231: ‘The analysis of the results showed a higher number of DCX+ cells in the hydrogel implanted with a delay compared to the group where the hydrogel is implanted immediately after the lesion, both at one week and one month after implantation (Figure 4)’. Please review/delete/modify as there are no statistical analysis provided to compare immediate versus delay so no critical evaluation of the data is possible.

 This reviewer finds the morphology of the GFAP staining hard to recognise as astrocytes in Figure 5. Please provide the data on which statistical analysis was done to allow the conclusion that ‘We did not see statistically significant differences between different groups) line 270 and please provide the representative images for one month after implant too (line 269).

Line 326: 'Quantification of the number of GFAP+ cells revealed that the number of astrocytes, in the delayed hydrogel implantation was higher within hydrogel implanted with a delay com- 327 pared to no delayed condition (7492±507 no delay and 6988±1155 with delay).'- please provide statistical analysis o back this up. Please also perform statistical analsysi for Figure 7D,K- if not significantly different please state this and do critical evaluation. Please explain why n numbers a re low and how animals were included/excluded. 

Line 326: 'Quantification of the number of GFAP+ cells revealed that the number of astrocytes, in the delayed hydrogel implantation was higher within hydrogel implanted with a delay compared to no delayed condition (7492±507 no delay and 6988±1155 with delay).' please modify as no statistical analysis was performed on figure 4. 

Line 385: 'One month after the surgery, we found 384 that the thickness of the glial scar was significantly decreased in the hydrogel implanted groups (Lesion: 34±3.9 µm; No delay: 26µm ±5.3; Delay: 20µm ±1; Figure 10 ).' please review as this is not significant according to Fig 10. 

Line 404- for analysis of microglia please provide the images and statistical analysis or delete this paragraph. 

For section 3.6 'Astrocytes and Microglia/Macrophage Polarization After hydrogel implantation' please provide statistical analysis for this section or delete as critical evaluation of the data is not possible especially given the data is SEM and no n numbers are provided. 

There are many conclusions and statements that are not backed up by the data, please review and modify accordingly.

Minor:

Please include details of methods in the abstract

please delete from line 77 in introduction: ‘The introduction should briefly place the study in a broad context and highlight why it is important. It should define the purpose of the work and its significance. The current state of the research field should be carefully reviewed and key publications cited. Please highlight controversial and diverging hypotheses when necessary. Finally, briefly mention the main aim of the work and highlight the principal conclusions. As far as possible, please keep the introduction comprehensible to scientists outside your particular field of research. References should be numbered in order of appearance and indicated by a numeral or numerals in square brackets—e.g., [1] or [2,3], or [4–6]. See the end of the document for further details on references.’

Please delete/reduce the results and conclusions of the study from the introduction

Methods line 94 please rewrite: ‘In no, delay group, due to a lack of tissue cavitation immediately after TBI, hydrogel was injected into the center of cortical injury. In delayed group, hydrogel was either injected or implanted after 20 min after preparation.’ Do you mean the no delay group received non-gelled form and the delay group received gel form? Please clarify

Reviewer 2 Report

The manuscript by Laine et al., investiagtes the therapeutic effect of biomaterial (hyaluronan-based) on host tissue after TBI in adult mice. They suggest that the biomaterial has a beneficial effect after TBI.

However, there are several issues that needs to be addressed before acceptance.

Introduction has some seemly random statements: 

Line 77-86. "The introduction should briefly place the study in a broad context and highlight why it is important. ...... See the end of the document for further details 85 on references." Please fix.

In the discussion:

Please add a paragraph or so about the limitations of the study. for example, the study does not investigate the behavioral changes in the animals. 

Recommend adding a few sentences about the future studies.

In the results: Some of the figures do not have any quantification. Is there any specific reason?

Why was a 2-way ANOVA used to compare two groups - as mentioned in the figure legend, for example: fig 7, 8.  However, the methods (statistical analysis) mentions one-way ANOVA. Please verify and clarify the correct statistical test is used for all the experimental analysis.

I'm confused by line 523 "6. Patents" 

Please review the manuscript for errors. 

Reviewer 3 Report

The manuscript by Laine et al. examined the therapeutic role of a biomaterial, the hyaluronan-based hydrogel, in injured brain tissue following TBI. They implanted the hydrogel into the motor cortex of adult mice with or without delay after lesion and examined cortical tissue loss, angiogenesis, neurogenesis and migration, glial scar formation, microglia activation, and neuroinflammation at one week and one month after the implantation. They demonstrated a beneficial role of the biomaterial in brain tissue repair following TBI. The manuscript contains multiple major and minor issues as listed below.

Introduction:

1.     Line 41: “NPCs neural progenitor cells (NPCs)” should be changed to “neural progenitor cells (NPCs)”

2.     The content between line 77, starting from “The”, and line 86 is obviously the Author Guide part. Why is it included here? 

3.     The related references should be cited for the two sentences in line 142-145.

Materials and Methods:

1.     A diagram illustrating the experimental design should be included to help readers better understand their experiments for different groups, hydrogel implantation time, and endpoint studies.

2.     There are numerous typos:

1.     All of the numbers, such as “0,5” and “2,5” shown in line 112, “1,5” in line 114, and those appeared in many other places across the whole manuscript should be changed to “0.5”, “2.5”, “1.5” etc...

2.     Line 137, “:” between “label” and “blood” should be deleted.

3.     Line 167, the last “***” should be “****”.

Results

1.     Lines 177-178, authors stated that “In delayed hydrogel implanted group, the size of lesion cavity is smaller than in other experimental group (p<0,0001).” However, their data actually showed an opposite result. Moreover, the authors need to specify the “other experimental group”.

2.     The statement (line 202-204) “the hydrogel CD31 density, increased and this increase was much more pronounced for no delayed implanted hydrogel compared to delayed implanted hydrogel” was not supported by their data.

3.     Fig 2E does not agree with the representative images shown in Fig 2A-2D.

4.     There is a typo in line 215, “they they”.

5.     There are no mention of Fig 3A’, B’ and C’ in the Results section, despite the images shown. 

6.     No p values in Fig 3D; no p values in Fig 4E, F, despite so called a significant difference mentioned in the text.

7.     Figure legend for Fig 4 mentioned “red” GFAP and “red” CD31 stained cells but there was no any red stained color shown in the images.

8.     Quantitative data should be included in Fig 5.

9.     The sentence in Lines 310-311 contains both grammatical issues and typo that need correction.

10.  In Lines 311-315, authors claimed that “the number of oligodendrocytes was higher within hydrogel implanted with a delay compared to no delay condition”, but this was not supported by their data. They also mentioned that they “observed 944.1±243.2 oligodendrocytes/mm² and 1.854±734.6 oligodendrocytes/mm² respectively, for hydrogel implanted immediately or with a delay”, indicating an opposite result as to what they claimed in the previous sentence. It is also unclear whether these results were from the one-week or one-month post-lesion. If they were from the one-week experimental mice, no p-value was shown between the two groups in Fig 6. There was no mention of one-year results.

11.  Similar problems also existed in Lines 326-328: although they stated that “delayed hydrogel implantation” showed a higher number of GFAP+ cells compared to the “no delayed” group, the numeric data they provided were just opposite to their statement. The data, 7292 from no delay group and 6988 from delay group did not agree with Figs. 7D, 7K either.

12.  Similarly, the statement in Lines 341-342 were not supported by the data shown in Fig 4.

13.  Line 244, Fig 8 should be changed to Fig 9, as no DCX/BrDU double labeling in Fig 8.

14.  Lines 353-354, Fig 9 should be changed to Fig 8.

15.  Line 357, Fig 8 should be changed to Fig 9.

16.  P-values should be included in the graphs shown in Fig 7, 8, and 9.

17.  The sentence for line 376-379 needs a reference. 

18.  The claim between lines 384 and 386 was not supported by the data shown in Fig 10D; in fact, there was no significantly difference between any two groups as they demonstrated.

19.  Line 407-409, authors might want to say that the two numbers were from two groups but here they said both were from “without delay”. Including a graph for this part should be helpful to understand their results.  

20.  Quantitative data for A1, A2, M1, M2 phenotypes should be included in Figs 11, 12, 13.

Discussion

1.     Some data showed better therapeutic effect for the no-delay condition than the delayed condition, but others revealed different results. This should also be discussed.

2.     It is necessary to discuss why the delayed implantation shows better effect than the immediate implantation in some experiments as they claimed. 

Round 2

Reviewer 3 Report

Some of previously identified issues still remain

1. Lines 175-176 stated that "the percentage of cortical loss increased in no delay..." but their data showed the opposite.

2. Line 192, "Figure 3A, C and D" should be Figure 3A, C, and E.

Author Response

Dear Editor,

Please find attached the corrected version of our manuscript.

Best regards

Afsaneh gaillard
